# Multiparametric MRI Features of Breast Cancer Molecular Subtypes

**DOI:** 10.3390/medicina58121716

**Published:** 2022-11-23

**Authors:** Madalina Szep, Roxana Pintican, Bianca Boca, Andra Perja, Magdalena Duma, Diana Feier, Bogdan Fetica, Dan Eniu, Sorin Marian Dudea, Angelica Chiorean

**Affiliations:** Department of Radiology and Medical Imaging, “Iuliu Hatieganu” University of Medicine and Pharmacy, 400374 Cluj-Napoca, Romania

**Keywords:** molecular subtypes, breast cancer, multiparametric MRI, ADC, luminal

## Abstract

*Background and Objectives*: Breast cancer (BC) molecular subtypes have unique incidence, survival and response to therapy. There are five BC subtypes described by immunohistochemistry: luminal A, luminal B HER2 positive and HER2 negative, triple negative (TNBC) and HER2-enriched. Multiparametric breast MRI (magnetic resonance imaging) provides morphological and functional characteristics of breast tumours and is nowadays recommended in the preoperative setting. *Aim*: To evaluate the multiparametric MRI features (T2-WI, ADC values and DCE) of breast tumours along with breast density and background parenchymal enhancement (BPE) features among different BC molecular subtypes. *Materials and Methods*: This was a retrospective study which included 344 patients. All underwent multiparametric breast MRI (T2WI, ADC and DCE sequences) and features were extracted according to the latest BIRADS lexicon. The inter-reader agreement was assessed using the intraclass coefficient (ICC) between the ROI of ADC obtained from the two breast imagers (experienced and moderately experienced). *Results*: The study population was divided as follows: 89 (26%) with luminal A, 39 (11.5%) luminal B HER2 positive, 168 (48.5%) luminal B HER2 negative, 41 (12%) triple negative (TNBC) and 7 (2%) with HER2 enriched. Luminal A tumours were associated with special histology type, smallest tumour size and persistent kinetic curve (all *p*-values < 0.05). Luminal B HER2 negative tumours were associated with lowest ADC value (0.77 × 10^−3^ mm^2^/s^2^), which predicts the BC molecular subtype with an accuracy of 0.583. TNBC were associated with asymmetric and moderate/marked BPE, round/oval masses with circumscribed margins and rim enhancement (all *p*-values < 0.05). HER2 enriched BC were associated with the largest tumour size (mean 37.28 mm, *p*-value = 0.02). *Conclusions*: BC molecular subtypes can be associated with T2WI, ADC and DCE MRI features. ADC can help predict the luminal B HER2 negative cases.

## 1. Introduction

The limitations of traditional histological classification led to the development of a new molecular classification of breast cancer (BC), in early 2000 [1]. The quantitative global gene expression profiling (GEP) technique identified four intrinsic subtypes: (1) estrogen receptor (ER) positive -> luminal A; (2) ER negative -> HER2-enriched (Erb-B2 overexpression); (3) basal-like and (4) normal breast-like. However, the GEP technique was not economical for daily practice. Nowadays, immunohistochemistry (IHC) procedures, using protein expression have been employed as a method for BC subtyping in clinical practice [2]. The IHC describes five surrogates BC subtypes: (1) luminal A; (2) luminal B HER2 negatives; (3) luminal B HER2 positive; (4) HER2 positives non luminal (corresponding to Erb-B2 overexpression in the intrinsic type classification) and (5) triple negative (the basal-like type of intrinsic subtyping).

The BC molecular subtypes have unique incidence, survival and response to therapy [3,4,5]. The ER, progesterone receptor (PR) and HER2 (human epidermal growth factor receptor (2) status provide prognostic and predictive information [6]. In this regard, the luminal A subtype is the most indolent BC and has the best prognosis; the TNBC type has a poor prognosis and is often associated with genetic mutations (BRCA 1) and HER2 positive subtypes benefit from targeted anti-HER2 therapy.

Breast MRI (magnetic resonance imaging) is recommended for the preoperative staging and subsequent choice of appropriate therapy in women with BC [7]. It provides morphological and functional characteristics of the breast tumours. Previously published work has independently investigated the role of T2-weighted image (WI) characteristics, apparent diffusion coefficient (ADC) or dynamic contrast enhancement (DCE) patterns in the prediction of BC molecular subtypes [8,9,10,11]. Few studies have analyzed combined multiparametric MRI findings (T2WI and ADC) for association with specific BC subtypes [12]. Recently, the distribution and level of breast parenchymal enhancement (BPE) has been reported to be associated with BC subtypes, allowing an additional risk stratification and targeted screening tests [10].

Our aim was to evaluate the multiparametric MRI features (T2-WI, ADC values and DCE) of breast tumours along with BPE features among different BC molecular subtypes.

## 2. Materials and Methods

### 2.1. Study Population

This was a retrospective study approved by the Ethics Committee of Medimages Review Board (NR10/15092022, from 15 September 2022), and the need for written consent was waived.

Inclusion criteria were patients with BC (regardless of the disease stage), preoperative breast MRI with T2WI, DWI/ADC, DCE sequences and pathology reports who presented to our clinic (Medimages Breast Center) between January 2018 and March 2022.

Exclusion criteria were patients with inadequate or incomplete MRI images, pathology and immunohistochemistry reports (Figure 1).

### 2.2. MRI Acquisition and Features

All patients underwent 1.5 T MRI examinations performed in the prone position with a coil dedicated to breast imaging, using two MR machines (Siemens Magnetom Symphony TIM and Altea). The breast MRI protocol consisted of five sequences: (1) T1-WI turbo spin echo; (2) T2-WI turbo spin echo (TR  =  5000 ms; TE = 120 ms, flip angle 90°; in-plane resolution 0.6 mm × 0.6 mm; 85 slices; slice thickness 2 mm; (3) T2 turbo inversion recovery magnitude (TIRM); (4) a T1WI vibe fs dynamic sequence (TR  =  4.66 ms; TE  =  2.3 ms; slice thickness 1.3 mm, with precontrast and five phases after the contrast administration (0.2 mL/kg, 3 mL/s); (5) DWI echo planar imaging with five b factors (0, 200, 400, 600, and 800 s/mm^2^). The ADC maps were automatically calculated linearly by the method provided by the MRI vendor.

All morphological MRI features (breast density, BPE, T2-WI and DCE features) were reported by one radiologist (AC) with more than 15 years of experience in breast imaging, using the American College of Radiology BI-RADS lexicon (5th edition) [13]. Each breast with more than one breast lesion was included in the analysis only once; their highest assessment was used to guarantee statistical independence of each observation.

The BPE was recorded as symmetric or asymmetric, with a minimal, mild, moderate or marked level. The T2-WI features included shape and margins; enhancement patterns, distribution and intensity (including kinetic curves) were recorded for mass and non-mass separately.

ADC values were obtained by placing a standardized intratumoral ROI of 0.2 mm^2^ in the darkest area corresponding to tumour enhancement. Breast parenchyma, tumour necrosis, haemorrhage and fat were excluded. Two ADC measurements were performed, first by the same experienced radiologist (AC) and second, by a moderately experienced breast radiologist with 5 years’ experience (MS).

### 2.3. Pathology and Immunohistochemistry Data

Age and pathology data were reviewed from the medical records, including the histologic tumour type (no special type—NST or special types), and immunohistochemistry findings (ER, PR, HER2 status, ki-67% proliferation index).

BC molecular subtypes were assigned according to the latest St. Gallen International Expert Consensus (2013) classification system for IHC subtypes 2 (Table 1).

### 2.4. Statistical Analysis

Statistical analyses were performed using MedCalc software (version 19.2.6, Ostend, Belgium). The Mann–Whitney U test was used to compare the age, size and ADC values between one molecular BC subtype versus all the other subtypes (for example ADC value of Luminal A BCs versus ADC value of all the other BC molecular subtypes). To analyse associations between each BC molecular subtype, clinicopathological data and MRI findings, the Chi-square or Fisher’s exact test were used. Using binary logistic analyses, we performed univariate and multivariate analyses to demonstrate the association between the presence of TNBC and different factors such as shape/margins/enhancement pattern. Results are expressed as unadjusted/adjusted odds ratio. ROC curve with AUC was calculated for statistically significant parameters. *p* < 0.05 was considered statistically significant.

The inter-reader agreement was assessed using the intraclass coefficient (ICC) between the ROI of ADC obtained from the two breast imagers. The following were selected as standards for strength of agreement: 0.01–0.20 = slight; 0.21–0.40 = fair; 0.41–0.60 = moderate; 0.61–0.80 = substantial; 0.81–0.99 = almost perfect; 1.0 = perfect.

## 3. Results

The study population consisted of 344 patients with breast cancer, with a 47.8 year mean age (ranging from 24 to 77 years old), having BC tumours with a mean size of 26 mm (ranging from 4.2 to 115 mm). The BC were as follows: 89 (26%) with luminal A, 39 (11.5%) luminal B HER2 positive, 168 (48.5%) luminal B HER2 negative, 41 (12%) triple negative (TNBC) and 7 (2%) with HER2 enriched.

### 3.1. Associations between 5 BC Molecular Subtypes and MRI Features

Patients with luminal B HER2 positive were associated with younger age compared to the other BC subtypes (44.5 mean age, *p*-value = 0.02), while patients with luminal A were associated with older age (50.2 mean age, *p*-value < 0.01).

The majority of the patients had a no-special-type (NST) BC histology type, with special subtypes (such as papillary or mucinous) associated with luminal A (*p*-value < 0.01).

There was no association between breast density and BC molecular subtypes (all *p* > 0.05).

The BPE was asymmetric in 18 (43.9%) patients with TNBC (*p*-value = 0.000), associated with moderate/marked level in 21 (51.21%) cases. There were no other statistically significant differences in BPE symmetry or level for the other BC molecular subtypes.

The smallest tumours were associated with luminal A (*p*-value < 0.01), and the largest with the HER2 enriched masses (*p*-value = 0.002).

For mass-appearing BC tumours, the shape was oval or round for 23 (56%, *p*-value = 0.01) of TNBC, with associated circumscribed margins (14.6%, *p*-value < 0.01) compared to the other BC subtypes (Figure 2).

A heterogeneous or rim enhancement was associated with TNBC (*p*-value < 0.01) in 26 (63.4%) and 15 (36.5%) cases, respectively. No BC mass was found to have internal dark septations (Figure 3 and Figure 4).

For the non-mass enhancement appearance, there were no differences in the distribution, enhancement type or mean ADC values between BC molecular subtypes (all *p* > 0.05).

The majority of tumours had a “wash-out” (type 3) enhancement curve, but only luminal A tumours reached statistical significance. In addition, luminal A tumours were associated with a plateau (type 2) kinetic curve (*p*-value = 0.03) (Table 2).

The univariate and multivariate analysis remained statistically significant for TNBC (Table 3).

The lowest mean ADC values were obtained for luminal B HER2 negative tumours (0.77, *p*-value < 0.01). For a cut-off value of 0.88 × 10^−3^ mm^2^/s^2^, there was an AUC = 0.583 in predicting luminal B HER2 negative cases (*p*-value < 0.01), with a sensitivity of 79.17 (95% CI 72.2–85.0), specificity of 36.36 (95% CI 29.3–43.9), 54.3 positive predictive value and 64.6 negative predictive value (Figure 5 and Figure 6).

We obtained almost perfect agreement between the ADC ROIs of the two readers (intraclass correlation = 0.85–0.91, weighted kappa = 0.78, standard error = 0.03, 95% CI 0.717–0.843).

### 3.2. Associations between ER/PR Positive and Negative BC and MRI Features

We further divided the entire study population into two groups according to positive or negative immunohistochemistry ER/PR status.

There were no differences for BC pathology type or breast density between the two groups (all *p*-values > 0.05).

Asymmetric BPE was associated with ER/PR negative group (*p*-value = 0.000), with no differences regarding the BPE level (*p*-value = 0.26).

Oval or round masses with circumscribed margins were associated with ER/PR negative group, while irregular masses with irregular or spiculated margins were associated with ER/PR positive tumours (*p*-values = 0.02 and < 0.01). All BC masses tested negative for ER/PR were associated with heterogeneous or rim enhancement (*p*-value < 0.01), while ER/PR positive tumours displayed rim enhancement only in 11% of the cases.

There were no differences regarding the ADC values or non-mass characteristics (distribution of enhancement, enhancement type) between ER/PR negative and positive tumours (all *p* > 0.05).

The persistent (type 1) kinetic curve was observed only in ER/PR positive tumours (4.6%) without reaching statistical significance (*p*-value = 0.65) (Table 4).

## 4. Discussion

In the current study, we found that T2WI, ADC values and DCE-based MRI features of BC differ between BC molecular subtypes.

In the univariate analysis, we found associations between shape, margins and enhancement pattern and TNBC. Lesions with oval/round margins or lobulated margins have more than two times higher odds of being TNBC when compared with the ones with an irregular shape. Circumscribed margins increased the odds of a lesion to be TNBC by 9.7 times. Moreover, the odds of TNBC status were approximately 4.3-fold greater in lesions with rim enhancement compared with the ones with heterogenous pattern. In the multivariate analysis, circumscribed margins and rim enhancement remained independently associated with TNBC.

As for the association of MRI features and ER/PR BC, our univariate analysis showed that lesions with irregular shapes are more likely to be ER/PR positive. Likewise, lesions with circumscribed margins have higher odds of being ER/PR positive. However, in the multivariate analysis, only circumscribed margins proved to be independently associated with ER/PR positive status, having an adjusted OR = 6.57 with a *p*-value < 0.01.

Few studies have included all five BC molecular subtypes in their analysis, and most papers have focused on differentiating ER/PR positive from ER/PR negative tumours to achieve statistical significance. In addition, the largest study population had 187 patients, and most authors reported association between BC molecular subtypes and one or two MRI based features (either T2WI features, or DWI/ADC or DCE features) [8,9,10,11]. We included 344 patients divided first into five BC molecular subtypes, and then into ER/PR positive and negative groups and analysed T2WI, ADC values and DCE features (enhancement pattern and kinetic curves) together.

Our results were consistent with other studies, which reported mass lesion type (*p* < 0.001), and smooth margins for TNBC [14,15,16]. Additionally, rim enhancement was reported in 76.8–80% of cases with TNBC, and has been proposed as a prognostic MRI feature for identifying TNBC [17,18]. In our study population, 15 cases (36.58%) presented with rim enhancement and reached statistical significance (*p*-value < 0.01).

It has been reported that ER/PR positive tumours (including luminal A, luminal B) are not associated with specific MRI features [18,19]. We found that irregular shaped masses with non-circumscribed margins were associated with the ER/PR positive group (all *p*-values < 0.05), in agreement with authors who reported that spiculated margins on ultrasound or mammography are associated with luminal tumours [20,21].

The persistent enhancement pattern (type I) has been associated in two studies with TNBC cases 14, 18, including in the young female population (< 30 years old). In our study, no TNBC mass had persistent kinetic enhancement, and in addition, rim enhancement was significantly associated with TNBC, consistent with Dogan et al. [15]. This could be explained by the heterogeneous group of TNBC, including special histology such as medullary carcinoma [22]. Another explanation of this heterogeneity is familial BC related to specific genetic alterations, which includes some cases of TNBC associated with benign imaging features including kinetics [23,24].

HER2 enriched tumours were found to have a kinetic curve pattern of more frequent “wash-out” or fast early rapid enhancement on MRI [25]. Our study did not reach statistical significance due to the small number of patients (only seven). However, we observed that this is associated with the ER/PR negative group, which includes the HER2 enriched patients.

There are conflicting results regarding the ADC/DWI, with authors reporting higher values in TNBC, and authors finding no association between ADC and BC molecular subtypes [19,26,27]. We found that luminal B HER2 negative was associated with the lowest ADC value (0.77 × 10^−3^ mm^2^/s^2^), which further predicts the BC molecular subtype with an accuracy of 0.583. Furthermore, we obtained almost perfect agreement between the two readers ADC ROIs, which supports the repeatability and reproducibility of this fast and easy method compared to the more technical ADC histogram making.

BPE may play a role as an imaging bridge to molecular BC subtypes allowing an additional risk stratification in breast MRI. Luminal B HER2 negative tumours may predominate in mild BPE, and TNBC in patients with marked BPE [28,29,30]. We found that asymmetric and moderate/marked BPE was associated with TNBC, and further with ER/PR negative group. However, no significant association was present between moderate/marked BPE and HER2 status or basal tumours (TNBC subtype) [2] and further studies are needed to make a conclusion.

Recent papers use radiomics-based MRI and machine learning in order to distinguish different BC molecular subtypes [31,32,33,34]. Leithner et al. reported 81–89% accuracy in distinguishing luminal A from luminal B, luminal B from triple-negative, luminal B from all others, and HER2-enriched from all others [35]. However, these techniques are time-consuming and the lack of standardized protocols remains a critical issue preventing entry into clinical practice.

This study had some limitations. First, this was a retrospective study, with one experienced breast imager reader that was aware of the presence of BC. Second, it was a single-centre study, which may weaken the statistical power of the results. Third, we did not include some features such as high T2WI signal, or DWI patterns (homogeneous or heterogeneous), as they are more prone to inter-reader variability. Fourth, even though this is the largest study on BC molecular subtypes, a larger prospective study needs to be conducted to validate the present results.

As future directions, where breast MRI will be increasingly used preoperatively, it could help change or adapt the patient management by suggesting a particular molecular type and may even indicate further examination (such as genetic testing for TNBC).

## 5. Conclusions

We found that BC molecular subtypes can be associated with multiparametric MRI features, especially TNBC, and that ER/PR positive tumours differ from ER/PR negative cancers. ADC can help predict the luminal B HER2 negative cases and ADC values obtained by ROI method is still a reliable method.

## Figures and Tables

**Figure 1 medicina-58-01716-f001:**
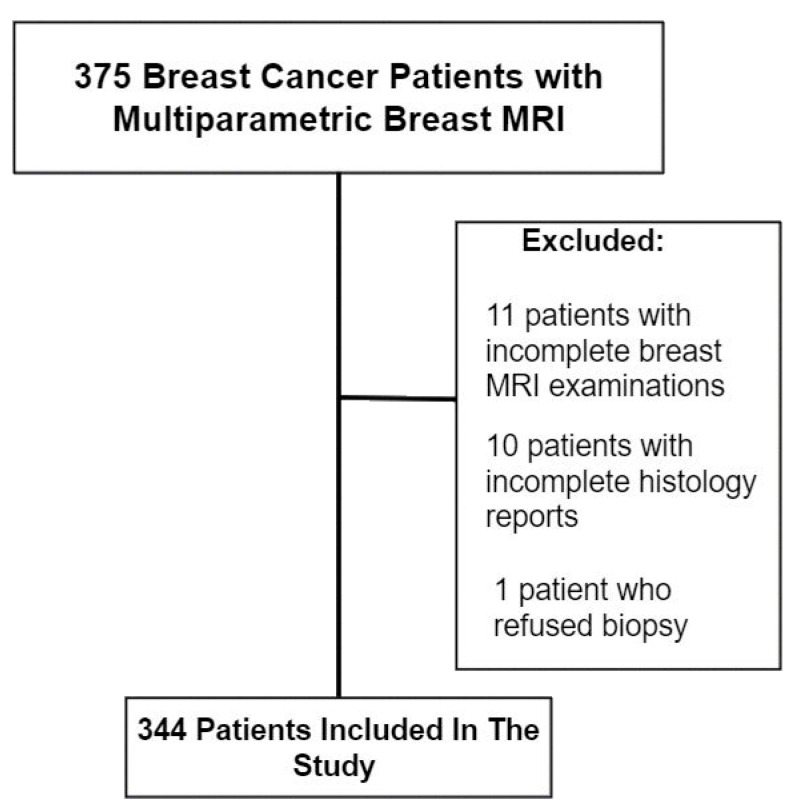
Study population.

**Figure 2 medicina-58-01716-f002:**
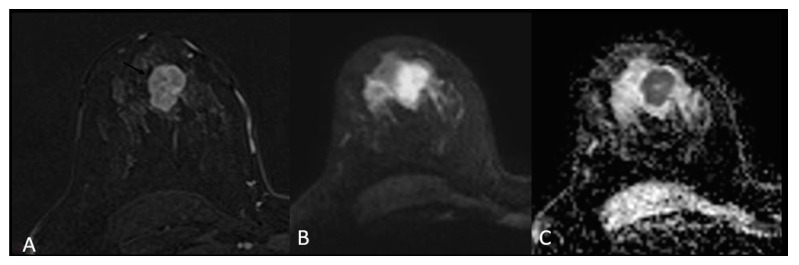
Triple negative breast cancer in a 33-year-old woman—subtracted T1-weighted contrast-enhanced image (**A**) shows a 3 cm oval (lobulated) mass with circumscribed margins, heterogeneous enhancement and restricted diffusion on DWI (**B**) and ADC (**C**).

**Figure 3 medicina-58-01716-f003:**
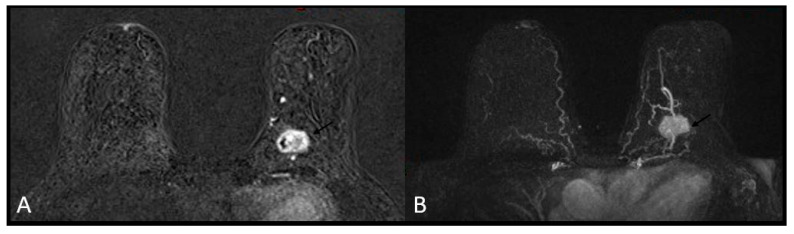
Triple negative breast cancer in a 60-year-old woman—heterogeneous enhancement on subtracted T1-weighted contrast-enhanced image (**A**) and maximum intensity projection (MIP, (**B**)).

**Figure 4 medicina-58-01716-f004:**
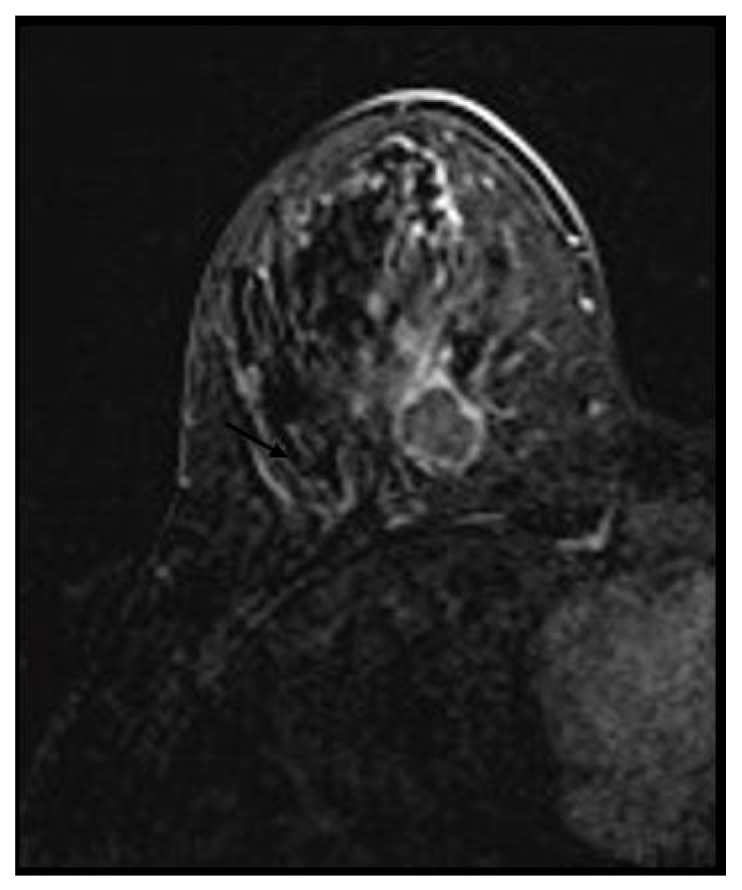
Triple negative breast cancer in a 32-year-old woman—rim enhancement on subtracted.

**Figure 5 medicina-58-01716-f005:**
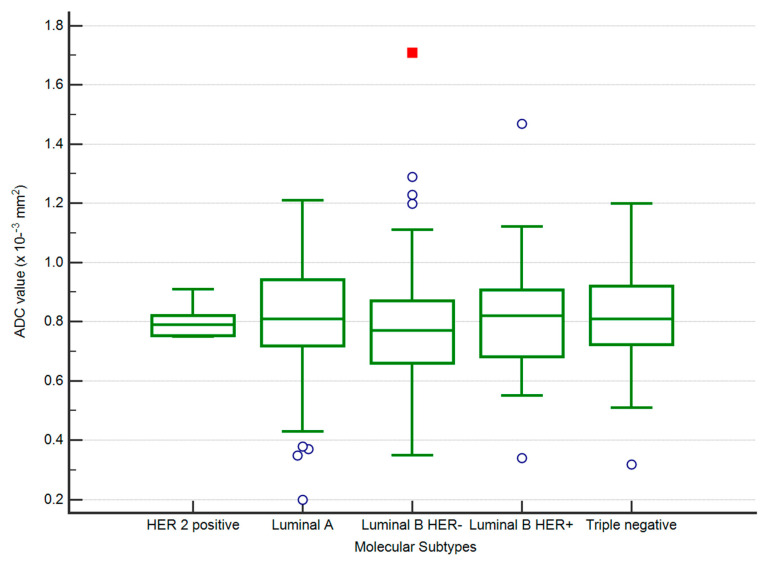
The distribution of ADC values in luminal B HER2 negative and other subtypes of BC.

**Figure 6 medicina-58-01716-f006:**
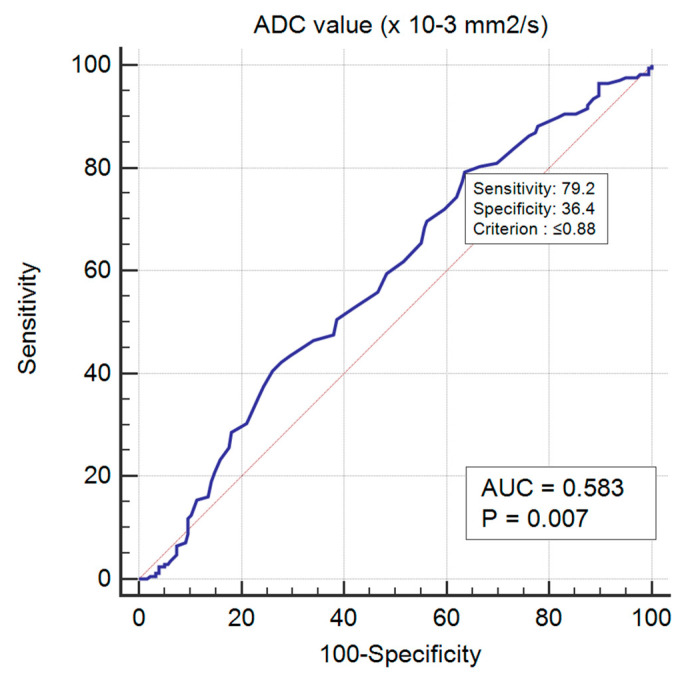
ROC curve for predicting luminal B HER2 negative BC.

**Table 1 medicina-58-01716-t001:** Breast cancer molecular subtypes based on immunohistochemistry findings.

Luminal A	Luminal B	Luminal B-like	HER2+	Triple Negative (TN)
ER+	ER+	ER+	ER−	ER−
PR+	HER2−	HER2+	PR−	PR−
HER2−	High Ki-67/PR−	Any PR	HER2+	HER2−
Low Ki-67		Any Ki-67		

ER = estrogen receptor, PR = progesterone receptor, HER2 = human epidermal growth factor 2, Ki-67% = proliferation index with low Ki-67% < 20% and high ki-67% > 20%.

**Table 2 medicina-58-01716-t002:** Breast cancer molecular subtypes—association between pathology and MRI features.

Variable	Luminal A	Luminal B HER+	Luminal B HER−	Triple Negative	Her 2 Positive
**Age mean**	50.2	44.5	48.2	45	44.4
*p*-value	<0.01	0.02	0.57	0.04	0.36
**Pathology**					
NST	58	33	140	36	0
Other	31	6	28	5	7
*p*-value	**<0.01**	0.41	0.09	0.16	0.35
**Breast density**					
A	6	1	6	3	0
B	17	6	39	9	2
C	45	21	78	16	3
D	21	11	45	13	2
*p*-value	0.55	0.67	0.68	0.60	0.90
**BPE**					

Symmetric	79	34	141	23	5
Asymmetric	10	5	27	18	2
*p*-value	0.05	0.36	0.35	**<0.01**	0.61

Level					
Minimal	22	5	36	13	1
Mild	19	6	35	7	0
Moderate	43	23	85	13	5
Marked	5	5	12	8	1
*p*-value	0.56	0.29	0.60	**<0.01**	0.46
**Mass**					

Size, mean	21.3	29.9	27.7	25.1	37.28
*p*-value	**<0.01**	0.13	0.09	0.59	**0.02**

Shape					
Oval	24	11	45	21	2
Round	18	5	17	2	2
Irregular	40	16	81	12	3
Lobulated	3	6	15	6	0
*p*-value	0.02	0.50	0.25	**<0.01**	0.58

Margins					
Circumscribed	0	1	4	6	0
Irregular	60	27	104	31	5
Spiculated	25	10	50	4	2
*p*-value	0.13	0.94	0.24	**<0.01**	0.88

Enhancement					
Homogeneous	2	0	2	0	0
Heterogeneous	75	33	138	26	5
Rim	8	5	18	15	2
*p*-value	0.17	0.73	0.28	**<0.01**	0.55

ADC mean	0.80	0.80	0.77	0.82	0.80
Range	0.20–1.21	0.34–1.47	0.35–1.71	0.75–0.91	0.32–1.2
*p*-value	0.16	0.54	**<0.01**	0.12	0.87
**Non-mass**					NA *

Enhancement					
Distribution					
Focal	1	0	4	0	
Lineal	2	1	4	2	
Segmental	3	0	7	0	
Regional	2	1	4	0	
M. regions	1	1	0	0	
Diffuse	1	0	0	0	
*p*-value	0.66	0.29	0.28	0.31	

Type					
Homogeneous	6	0	1	1	
Heterogenous	4	3	7	1	
Clumped	0	0	9	0	
Cluster ring	0	0	1	0	
*p*-value	0.82	0.51	0.37	0.99	

ADC mean	0.88	0.86	0.86	NA *	
Range	0.74–1.02	1 case	0.60–1.29		
*p*-value	0.48	0.81	0.58		
**Kinetic curves**					
Persistent (1)	6	2	5	0	0
Plateau (2)	22	7	30	9	3
Wash-out (3)	61	30	133	32	4
*p*-value	0.10	0.82	0.30	0.39	0.31
**ADC mean**	0.80	0.81	0.77	0.81	0.80
(mass + non-mass)					
Range	0.20–1.21	0.34–1.47	0.35–1.71	0.34–1.47	0.75–0.91
*p*-value	0.12	0.59	**<0.01**	0.15	0.88
**Total**	89	39	168	41	7

* NA—impossibility to perform statistical tests; no dark septations were observed for the mass enhancement type; size is displayed in mm; range = minimum and maximum values.

**Table 3 medicina-58-01716-t003:** Odds ratio for TNBC on univariate and multivariate analysis.

Variable	Odds Ratio (95%CI)	*p*-Value
**Univariate analysis**		
Shape − oval + round	2.16 (1.03–4.52)	0.04
Shape − lobulated	2.91 (0.99–8.51)	0.05
Margins − circumscribed	9.70 (2.81–33.45)	< 0.01
Enhancement − homogenous	0.00	0.99
**Multivariate analysis**		
Shape – oval + round	1.43 (0.64–3.19)	0.39
Shape − lobulated	2.64 (0.88–7.92)	0.08
Margins − circumscribed	9.12 (2.19–37.87)	< 0.01
Enhancement − homogenous	0.00	0.99

**Table 4 medicina-58-01716-t004:** ER/PR positive and ER/PR negative cancers in association with MRI features.

Variable	ER/PR Positive	ER/PR Negative	*p*-Value
**Pathology**			0.06
NST	231	43
Other	65	5
**Density**			0.67
A	13	3
B	62	11
C	144	19
D	77	15
**BPE**			**<0.01**

Symmetric	254	28
Asymmetric	42	20

Minimal	63	14	0.26
Mild	60	7
Moderate	151	18
Marked	22	9
**Mass**			

Size	26.08	26.95	0.14

Shape			**0.02**
Oval	80	23
Round	40	4
Irregular	137	15
Lobulated	24	6

Margins			**<0.01**
Circumscribed	5	6
Irregular	191	36
Spiculated	85	6

Enhancement			**<0.01**
Homogeneous	4	0
Heterogeneous	246	31
Rim	31	17

ADC mean	0.82	0.78	0.16
**Non-mass**			

Enhancement			0.31
Distribution		
Focal	5	0
Lineal	7	2
Segmental	10	0
Regional	7	0
M. regions	2	0
Diffuse	1	0

Enhancement			0.95
Homogeneous	16	1
Heterogenous	13	1
Clumped	2	0
Cluster ring	1	0
**Kinetic curves**			0.26
Persistent	13	0
Plateau	59	12
Wash-out	224	36
**Total**	281	48

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
