# Peer review of "Multiparametric MRI Features of Breast Cancer Molecular Subtypes"

_medicina, 2022, doi:10.3390/medicina58121716_

Round 1

Reviewer 1 Report

In the article titled “Multiparametric MRI features of breast cancer molecular sub- 2

Types”, Dr.Szep et al. reported multiparametric MRI evaluation results of breast cancer and tried to explore the associations of specific imaging characteristics with the molecular subtypes of breast cancer.

The topic may be of interest to radiologists and oncologists, and this is a retrospective study with relatively large sample size. I am not a radiologist, it is not my position to judge the part of “MRI acquisition and features”, However, I find the methodology of the study, particularly the statistical analysis, problematic.

1.       The data presentation is insufficient (e.g., range or SD should be presented in addition to mean), the original tabulated data should be presented as supplementary data.

2.       Mann-Whitney U test is a nonparametric test for data comparison between two independent groups, the usage of this method in this article is inappropriate, it is unknown how the p values (for age and ADC) were calculated. An ANOVA or Kruskal-Wallis test should be considered, and only one p value should be reported for age and for ADC, respectively.

3.       The Chi-square and Fisher’s exact tests were incorrectly conducted, it is unknown how the p values were calculated.

4.       The authors’ conclusion “Patients with TNBC are more likely to have round or oval shaped breast masses with circumscribed margins, and will more frequently show heterogeneous or rim enhancement on DCE sequences.” may not be necessarily wrong, but the clinical significance of this statement is fairly low, it will be more meaningful if the author can prove oval/round + circumscribed margin + heterogeneous/rim masses are more likely to be TNBC, calculate the relative risk /odds ratio.

5.       The conclusion “We found that irregular shaped masses with non-circumscribed margins were associated with ER/PR positive group (all p-values < 0.05)”, same as the above comment, the relative risk/odds ratio should be calculated in order to prove this conclusion.

6.       The ROC curve for ADC value: The distribution of ADC in luminal B HER2- and the other subtypes should be presented as scattered/box plots. There have been many other studies reporting MRI ADC in luminal B HER- subtype (PMID: 31690273, 34345946) where negative associations were reported, please discuss the discrepancy between the current study and previous studies.

Reviewer 2 Report

In this manuscript, the authors performed a retrospective study based on all underwent multiparametric breast MRI (T2WI, ADC and DCE sequences) and extracted thier features according to the latest BIRADS lexicon. Then, they explored the association between molecular subtypes of multiparametric MRI features of breast cancer and the classfic typing based on surface receptor of breast cancer cells, and found that there are some correlation between two classification categories. In a word, this manuscript provide a new classification method based on molecular phenotype of breast cancer,  which would help to clinical diagnosis and treatment of breast cancer. But I have several following concerns:

1) Line 23, the "p" of "p values"should be in italics. please also double check the similar errors in the text.

2) Line 24, "0.77 x 10-3mm2/s2", the "-3", "2" and "2" should be superscript. please also revise the similar format errors in the text.

3) The abbreviations in the text should be defined the first time they appear. such as "MRI", "HER2"...

4) Figure 2 is a table pasted from another article. We suggest that the author redraw the standard three-line table or apply for copyright.

5) Please add a scale bar for the Figures 3-5 in the text.

6) Please revise all the tables in the text into standard three-line tables.

7) Please specify each author's contribution in "Author Contributions" section in the text.

8) Fund information should also be added.

9) The format of references should be uniform. It is recommended to add DOI numbers of Refferences.

10) There are also spelling or formatting errors in the text, the auhtors would better double checked all the text and revised them.

Reviewer 3 Report

The manuscript “Multiparametric MRI features of breast cancer molecular subtypes” met its primary objective of identifying certain MRI characteristics associated with specific breast cancer subtypes. The use of MRI as part of the preoperative evaluation in breast cancer has increased in daily practice, therefore, the studying the predictive/prognostic value of certain MRI features has become particularly relevant. Before this manuscript is suitable for publication I would recommend the following modifications:

General recommendations:

- Thorough review of complete manuscript to correct any mistakes regarding spelling and grammar is needed.

- I would recommend joining paragraphs with same general idea. A lot of paragraphs are just isolated sentences.

Introduction

- The paragraphs within the introduction lack cohesion and unity between ideas presented. Please use more connectors.

- The first two paragraphs could be joined into a single paragraph. Maybe mention and cite that the St Gallen consensus recommends using IHC as a surrogate form for classifying molecular subtypes.

- The third paragraph lacks further information about value of such subtypes, for example, triple negative has worse prognosis, luminal are more frequent, HER2 benefit from targeted therapy, etc.

- For the paragraph: “Recently, the distribution and level of breast parenchymal enhancement (BPE) has been reported to be associated with BC subtypes, allowing an additional risk stratification and targeted screening tests.” Reference is missing and could be put together with the previous paragraph describing other prior studies evaluating MRI characteristics.

- Aim should be described in past tense.

Methods

- The authors should use the STROBE guidelines for observational studies to report methods and results in this manuscript. I recommend re-structuring the methods section to cover at least: study design, setting (specifying where the study took place), participants (were patients with metastatic disease excluded?), variables (clinical, pathological, and MRI features), and statistical methods.

- Authors should declare if this study received any type of IRB approval.

- Authors should specify how clinical and pathological information was collected (for example: from medical records?)

- Add the initials of the radiologist to this sentence: “All morphological MRI features (breast density, BPE, T2-WI and DCE features) were 82 reported by one radiologist with more than 15 years of experience in breast imaging, using 83 the American College of Radiology BI-RADS lexicon (5th edition) 13”

Results

Again, the authors should follow the STROBE guidelines for reporting the results in this manuscript.

- The authors should first describe the general clinical characteristics of the patients included in this study which were later analyzed for associations for example: age, histologic subtype, breast density, tumor size, etc. I would recommend adding this information to the first paragraph of the results section.

- Table 1 should be referenced earlier on the manuscript as the associations between age, histologic subtype, breast density, tumor size, etc., are described in the second subsection of results.

- If available, it would be interesting to include some other clinical and pathological characteristics such as menopausal state, histologic grade, tumor size, nodal status, and clinical stage.  

- P-values of 0.000 should be reported as <0.01 or <0.001. Authors should be consistent on the number of decimal points used within the manuscript (please choose between 2 or 3 digits, I recommend using 2 digits) as some p values are reported with 3 digits, for example: p = 0.024 (for shape in Table 2) and some with 2 digits p = 0.26 (for marked BPE also in Table 2).

Discussion

- Again, paragraphs discussing the same main idea should be merged to have more complete paragraphs and cohesion within the text. The authors should also discuss the findings of this study in the same order as they were presented in the results section.

- Limitations should include that the radiologists evaluating MRI features are highly experienced, maybe less experienced radiologists may not be able to detect subtle differences or inter-observer variability could be higher.

- As stated as a limitation, authors should clarify why some features such as high T2WI signal, or DWI patterns (homogeneous or heterogeneous) were not evaluated in this analysis.

- Authors should also further discuss the importance or future implications of identifying/detecting MRI characteristics associated with certain molecular subtypes, for example, will patients be able to forego diagnostic biopsies? What is the practical application?
